# PAGER: A FRAMEWORK FOR FAILURE ANALYSIS OF DEEP REGRESSION MODELS

## ABSTRACT

Safe deployment of AI models requires proactive detection of potential prediction failures to prevent costly errors. While failure detection in classification problems has received significant attention, characterizing failure modes in regression tasks is more complicated and less explored. Existing approaches rely on epistemic uncertainties or feature inconsistency with the training distribution to characterize model risk. However, we find that uncertainties are necessary but insufficient to accurately characterize failure, owing to the various sources of error. In this paper, we propose PAGER (Principled Analysis of Generalization Errors in Regressors), a framework to systematically detect and characterize failures in deep regression models. Built upon the recently proposed idea of anchoring in deep models, PAGER unifies both epistemic uncertainties and complementary non-conformity scores to organize samples into different risk regimes, thereby providing a comprehensive analysis of model errors. Our results highlight the capability of PAGER to identify regions of accurate generalization and detect failure cases in out-of-distribution and out-of-support scenarios.

## 1 INTRODUCTION

An important aspect of safe AI model deployment is to proactively detect potential failure modes to enable practitioners avoid costly errors. In classification tasks, this is often posed as generalization gap prediction, where the goal is to estimate the expected deviation in model accuracy between an unlabeled test set and a controlled validation set (Guillory et al., 2021; Narayanaswamy et al., 2022; Baek et al., 2022). Instead, our focus in this paper is on failure detection with deep regression models, motivated by their prominence in several critical applications including healthcare (Luo et al., 2022; Young et al., 2020), autonomous driving (Huang & Chen, 2020), and physical sciences (Raissi et al., 2019). In general, characterizing failure modes in continuous-valued prediction tasks is more complex, since the notion of failure is subjective and error tolerances can vary across different use cases. Consequently, this problem has not been sufficiently explored until recently.

Most commonly, epistemic uncertainties (Lakshminarayanan et al., 2017; Gal & Ghahramani, 2016; He et al., 2020; Amini et al., 2020) have been considered to be a reasonable surrogate for expected risk (Lahlou et al., 2023). However, in practice, failure detection performance using uncertainty alone can be poor as low uncertainty regimes can still correspond to a higher risk due to feature heterogeneity in the training data (Seedat et al., 2022) or, data regimes outside the training support may correspond to low risk if the model extrapolates accurately. Figure 1 illustrates the lack of strong correlation between uncertainty and the true risk using a simple 1D function (with two different experiment designs). On the other hand, DataSUITE (Seedat et al., 2022) recently proposed to qualify failure modes solely based on feature inconsistency with respect to the training distribution (using an auto-encoding error). Since this approach is task-agnostic by design, its characterization can be limiting for arbitrary target functions.

In this paper, we introduce PAGER (Principled Analysis of Generalization Errors in Regressors), a new framework for failure characterization in deep regression models. At the outset, our approach proposes to move away from sample-level analysis to identifying groups of varying expected risks. More specifically, we organize samples from a test set into *ID* (i.e., in distribution, where we expect the model to generalize), *Low Risk*, *Moderate Risk* and *High Risk* regimes, thus enabling a comprehensive analysis of model errors. Given the inherent insufficiency of using only uncertainties, PAGER

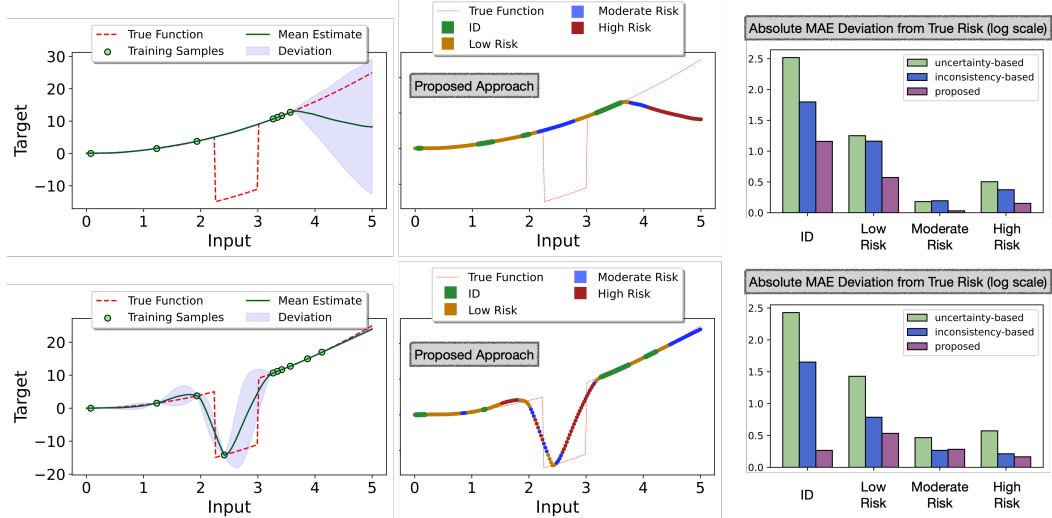

Figure 1: **Epistemic uncertainty, while necessary, is not sufficient to completely characterize all risk regimes.** Top: Out-of-support (OOS) samples in the range of $[2.2 - 2.7]$ exhibit low uncertainty but moderate risk due to significant deviation from true function. Bottom: Even with better experiment designs, uncertainty alone in the extrapolating regime $[4.5 - 5]$ is unreliable due to potential drift from the truth. We propose PAGER, a framework that leverages anchoring (Thiagarajan et al., 2022) to unify prediction uncertainty and non-conformity to the training data manifold. PAGER accurately flags those erroneous regimes as Moderate Risk (shown in blue) and outperforms existing baselines in accurately categorizing samples consistent with the true risk (lower MAE).

estimates both epistemic uncertainties and novel non-conformity scores that measure adherence to the training data manifold, using a unified anchoring-based approach (Thiagarajan et al., 2022; Netanyahu et al., 2023). For the examples in Figure 1, we show the difference between the true risk and the predicted risk in each of the four regimes. When compared to state-of-the-art uncertainty-based and inconsistency-based detectors, we find that, the risk regimes identified by PAGER effectively align with the true risk. Finally, we advocate for a suite of metrics that can holistically assess failure detectors in regression tasks, and perform empirical studies with both tabular data and image regression benchmarks. Our results show that, PAGER accurately detects failures in both out-of-distribution and out-of-support settings, while also identifying regions of accurate generalization.

## 2 BACKGROUND AND RELATED WORK

**Preliminaries.** We consider a predictive model $F_\theta$, parameterized by $\theta$, trained on a labeled dataset $\mathcal{D} = \{(x_i, y_i)\}_{i=1}^M$ with $M$ samples. Note, each input $x_i \in X$ and label $y_i \in y$ belong to the spaces of inputs $X$ (in $d$−dimensions) and continuous-valued targets $y$ respectively. Given a non-negative loss function $\mathcal{L}$, e.g., absolute error $|y - \hat{y}|$, the sample-level risk of a predictor can be defined as $R(x; F_\theta) = \mathbb{E}_{y|x} \mathcal{L}(y, F_\theta(x))$. Basically, risk is defined as the cost incurred for incorrect predictions. While estimating true risk is challenging in practice due to the need for access to the unknown joint distribution $P(X, y)$, it becomes crucial to develop methods that can reliably flag and categorize different risk regimes to facilitate safe deployment of models. We now define the different regimes of generalization that we want to characterize: (i) *In-distribution*: This is the scenario where $P(x_t \in X) > 0$ and $P(x_t \in \mathcal{D}) > 0$, i.e., there is likelihood for observing the test sample in the training dataset; (ii) *Out-of-Support* (OOS): The scenario where $P(x_t \in X) > 0$ but $P(x_t \in \mathcal{D}) = 0$, i.e., the train and test sets have different supports, even though they are drawn from the same space; (iii) *Out-of-Distribution* (OOD): This is the scenario where $P(x_t \in X) = 0$, i.e., the input spaces for train and test data are disjoint. We illustrate the differences between OOS and OOD regimes in Figure 2 using 1D and 2D examples. In the 1D case, OOS corresponds to regimes where the likelihood of observing data in the training support is zero but is non-zero in the input-space. Data from regimes

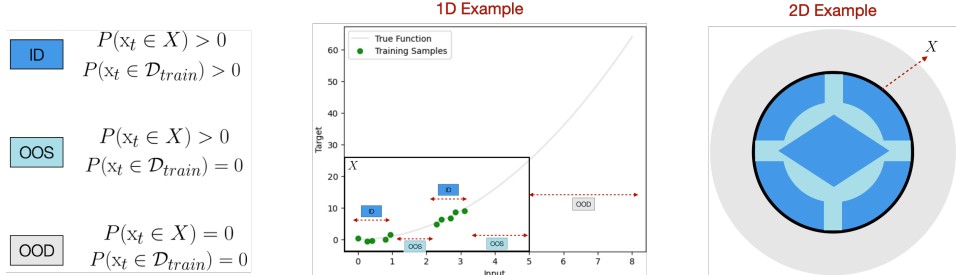

Figure 2: **An illustration of different data regimes of generalization.** Using examples in 1D and 2D, we show ID, OOS and OOD regimes.

outside the input space are referred to as OOD. In the 2D case, OOS constitutes regimes with new combinations of features (light blue) which are not jointly but individually seen in the train data.

**Failure Characterization**. Generalization gap predictors estimate the difference in accuracy between an unlabeled, distribution shifted dataset with respect to a controlled validation data. Focused on classification, these methods estimate either sample level *correctness* (Ng et al., 2022; Jiang et al., 2022) or distribution-level metrics (Guillory et al., 2021; Narayanaswamy et al., 2022; Chen et al., 2021; Jiang et al., 2019; Deng & Zheng, 2021) to estimate the gap. Risk estimation in regression models is a relatively under-explored area of research. Among existing methods, DEUP (Lahlou et al., 2023) is a recent approach that utilized predictive uncertainty as a surrogate for total risk, which we illustrated to be insufficient for failure detection in 1. Conformal prediction (CP) forms another popular class of uncertainty estimation methods (Vovk et al., 2005; Lei et al., 2018), that can be leveraged to identify risk regimes. However, with OOS and OOD data, the exchangability assumption made by conventional CP frameworks is violated (Tibshirani et al., 2019) and hence the estimated intervals are ineffective in our setting. Consequently, CP-based methods like DataSUITE (Seedat et al., 2022) handle this by considering only the input domain (and not the target) to construct non-conformity scores. However, our results show that it is incapable of identifying errors in OOS regimes. Instead, for the first time, we find that a combination of epistemic uncertainty along with the proposed non-conformity scores strongly correlate with risk in all regimes. Note, PAGER does not transform the non-conformity into intervals, and estimates the score even for test data unlike CP.

**Anchoring in Predictive Models.** The anchoring principle involves reparameterizing an input sample x (referred to as the *query*) into a tuple comprising an *anchor* r drawn from the training distribution and the residual $\Delta$x denoted by $[r, \Delta x] = [r, x - r]$ (Thiagarajan et al., 2022). It thus induces a joint distribution that depends not only on $P(X)$, but also on the distribution of residuals $P(\Delta)$. During training, anchoring ensures consistency in prediction for a query x by effectively modeling the combinatorial relationship between every sample in the dataset and infers the joint distribution $P(X, \Delta)$. During inference, we can obtain accurate predictions for a query $x_t$ if $x_t \in P(X)$ and $[x_t - r] \in P(\Delta)$. This idea has shown promising results in various tasks (Thiagarajan et al., 2022; Narayanaswamy et al., 2022; Trivedi et al., 2023; Netanyahu et al., 2023) including uncertainty estimation, anomaly detection and extrapolation. Our framework PAGER makes an interesting finding that both uncertainty and manifold non-conformity to the training manifold, two key components of failure characterization, can be estimated using the anchoring principle. We provide the detailed description of anchoring based methods in Appendix A.1 of the supplementary material.

## 3 CHARACTERIZING AND DETECTING FAILURE IN DEEP REGRESSORS

In this paper, we propose a novel framework for systematically characterizing failure in deep regression models. This framework organizes unlabeled samples from a test set into different regimes based on their levels of expected risk (*ID, low, moderate, high*). By doing so, practitioners can gain a detailed understanding of a model's generalization behavior. Next, we make a critical advancement to the challenging problem of estimating sample-level risk. Existing approaches utilize epistemic uncertainties or task-agnostic data inconsistency to define surrogate measures for expected risk. In contrast, our method leverages the principle of neural network anchoring (Thiagarajan et al., 2022; Netanyahu et al., 2023) to unify both prediction uncertainty and manifold non-conformity (MNC) to

the training data manifold, which are subsequently used to derive the risk regimes. In addition to eliminating the need for separate estimators for uncertainties and the proposed scores, our approach does not require additional calibration data. Finally, we introduce a suite of evaluation metrics to quantitatively benchmark failure detectors in deep regression models.

## 3.1 MEASURING UNCERTAINTY AND MANIFOLD NON-CONFORMITY VIA ANCHORING

The central idea of our approach is to not only accurately estimate the uncertainty for a sample, but also its (non-)conformity to the training data manifold. This is motivated by the observation that, regardless of the uncertainty, a model can induce a large error for test sample $x_t$, when $(x_t, y_t) \notin P(X, y)$, i.e., , the risk can be high when the sample does not adhere to the data manifold. While there are numerous options available for estimating epistemic uncertainty in deep models (Gawlikowski et al., 2023; Yang et al., 2021), measuring non-conformity without ground truth is not straightforward. Consequently, existing methods adopt simplified scoring functions only based on the input data (without labels) (Seedat et al., 2022) or utilize conformal prediction strategies to transform scores into well-calibrated intervals so that they need not be explicitly computed for test data (Teng et al., 2023). While the former approach does not leverage the characteristics of the task at hand, the latter is not applicable in our scenario due to the violation of exchangeability condition w.r.t OOS and OOD data regimes. Hence, we propose an alternative approach based on anchored neural networks.

**Uncertainty via forward anchoring:** An anchored model is trained by transforming a training sample x into a tuple, $[r, x - r]$ based on an anchor r, which is also drawn randomly from the training dataset $\mathcal{D}$. Building upon the findings from (Thiagarajan et al., 2022), at test time, predictions from different anchor choices can be used to obtain the mean and uncertainty estimates as follows:

$$\mu(y_t | x_t) = \frac{1}{K} \sum_{k=1}^{K} F_{\theta^*}([r_k, x_t - r_k]); \quad \sigma(y_t | x_t) = \sqrt{\frac{1}{K-1} \sum_{k=1}^{K} (F_{\theta^*}([r_k, x_t - r_k]) - \mu)^2}, \quad (1)$$

where $\mu$ and $\sigma$ are estimated by marginalizing across $K$ anchors $\{r_k\}_{k=1}^{K}$ sampled from $\mathcal{D}$.

**Non-conformity via reverse anchoring:** Turning our attention to the assessment of non-conformity, we make a noteworthy observation regarding the flexibility of an anchored neural network. It is able to not only capture the relative representation of a query (i.e., test sample) in relation to an anchor (i.e., training sample), but also the reverse scenario. To elaborate, the prediction for an anchor sample r is given as $F([x_t, r - x_t])$, where $x_t$ represents a test sample. Since the ground truth function value is known for the training samples, we can measure the non-conformity score for a query sample based on its ability to accurately recover the target of the anchor. Note, unlike existing approaches, this can be directly applied to unlabeled test samples and does not require explicit calibration.

Looking from another perspective, the original anchor-centric model ((Thiagarajan et al., 2022)) provides reliable predictions for an input $[r, \Delta]$ only when $r \in \mathcal{D}$ and $\Delta \in P(\Delta)$. However, for OOD or OOS samples, if $\Delta \notin P(\Delta)$, the estimated uncertainty becomes large everywhere, and as a result becomes inherently unreliable in order to rank by levels of expected risk. In contrast, our proposed query-centric score overcomes this challenge by directly measuring the discrepancy with respect to the ground truth target. Specifically, we define our non-conformity score as follows:

$$\texttt{Score}_1(x) = \max_{r \in \mathcal{D}} \left\| y_r - F([x, r - x]) \right\|_1 \quad (2)$$

It is important to note that we measure the largest discrepancy across the training dataset. In practice, this can be done for a small batch of randomly selected training samples (e.g., 100). As demonstrated in our results, our proposed non-conformity approach proves highly effective and efficient compared to state-of-the-art uncertainty-based and inconsistency-based failure detectors (refer to Figure 1).

**Resolving moderate and high risk regimes better:** A closer examination of equation 2 reveals that for samples that are far away from the training manifold, the model prediction can be uniformly bad (i.e., extrapolation), as both $x \notin \mathcal{D}$ and $\Delta \notin P(\Delta)$. This can make distinguishing between samples with moderate risk and those with high risk very challenging. To mitigate this situation, we propose to transform both the query x (used as the anchor in reverse anchoring) and $\Delta$ to be

in-distribution so that the anchored model $F$ can produce reliable predictions. We achieve this using the following optimization problem:

$$\texttt{Score}_2(\mathrm{x}) = \max_{\mathrm{r} \in \mathcal{D}} \left\| \mathrm{x} - \arg\min_{\bar{\mathrm{x}}} \left( \left\| \mathrm{y}_{\mathrm{r}} - F([\bar{\mathrm{x}}, \mathrm{r} - \bar{\mathrm{x}}]) \right\|_1 + \lambda \mathcal{R}(\bar{\mathrm{x}}) \right) \right\|_2,$$

$$\text{where } \mathcal{R}(\bar{\mathrm{x}}) = \left\| \bar{\mathrm{x}} - A([\mathrm{x}, \bar{\mathrm{x}} - \mathrm{x}]) \right\|_2 + \left\| \mathrm{x} - A([\bar{\mathrm{x}}, \mathrm{x} - \bar{\mathrm{x}}]) \right\|_2. \tag{3}$$

In this approach, the score is measured as the discrepancy in the input space to a new fictitious sample that serves as an intermediate anchor, such that its prediction matches the known prediction on the training sample. In other words, we optimize the modification of the query sample x to x̄ in such a way that we accurately match the true target for the anchor r. The non-conformity is then quantified as the amount of movement required in x to match the target. To ensure that the resulting x̄ remains within the input data manifold, we incorporate a regularizer $\mathcal{R}(\bar{\mathrm{x}})$. Specifically, we train an anchored auto-encoder $A$ on the training dataset $\mathcal{D}$ and enforce cyclical consistency, where $A$ is required to recover x using x̄ as the anchor and vice versa. While $\texttt{Score}_1$ is extremely scalable, $\texttt{Score}_2$ provides better resolution in the medium and high risk regimes at an increased compute cost. In practice, the choice of the non-conformity score can be based on the compute constraints and risk tolerance in an application. We provide the algorithm listings and details of all these models in Appendix A.2.

## 3.2 PAGER FRAMEWORK

Since it is challenging to accurately estimate and interpret sample-level error estimates, particularly in OOS or OOD regimes, a more tractable approach is to analyze sample groups that correspond to varying levels of expected risk. To this end, we develop our PAGER framework for deep regression models (Figure 3). At a high level, we organize the set of test samples based on both uncertainty and MNC. Without loss of generality, we assume that a typical test set contains samples close to the training distribution, as well as OOS and potential OOD samples. Note, even when this assumption does not hold and the test set does not contain distribution shifts, our proposed framework can still identify regimes with increasing levels of expected risk.

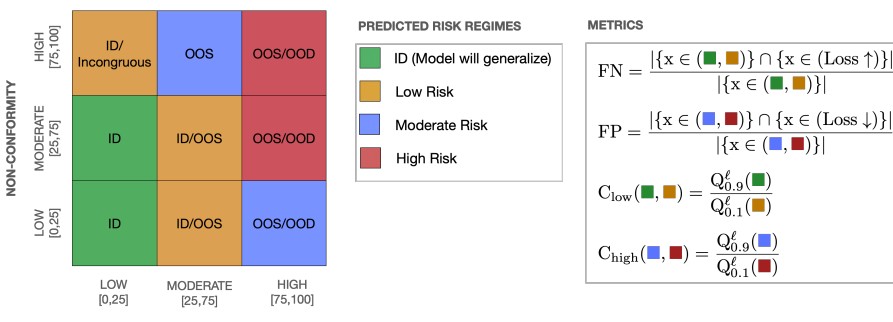

Figure 3: **Overview of our proposed framework.** PAGER organizes test examples into bins (*low*, *moderate* and *high*) using both predictive uncertainty and MNC scores. With such a categorization, PAGER associates samples into 4 levels of expected risk (ID, Low Risk, Moderate Risk and High Risk). We also advocate a suite of metrics that enables a holistic assessment of failure detectors.

In our implementation, both scores are split into three bins using conditional quantile ranges (*low*:$[0, 25]$, *moderate*:$[25, 75]$ and *high*:$[75, 100]$), thereby creating a non-trivial partition of the test data into risk regimes. Note that, the number of bins and the threshold choices can be adapted to the specific application, and can even be selected using an additional calibration dataset. However, we emphasize that, a vanilla implementation of PAGER does not require any calibration step and can directly work on the unlabeled test set. We now describe the different risk regimes in PAGER.

**ID** (■): The model generalizes well in this regime and is expected to produce low prediction error. In our framework, this corresponds to samples with low uncertainty and low/moderate MNC scores;

**Low Risk** (■): Even when the uncertainty is low, the model can produce higher error than the ID samples, when there is incongruity (e.g., samples within a neighborhood having different target

values). Similarly, for OOS samples with moderate uncertainties, the model can still extrapolate well and produce reduced risk. Hence, we define this regime as the collection of (low uncertainty, high MNC) and (moderate uncertainty, low/moderate MNC) samples;

**Moderate Risk (■):** Since epistemic uncertainties can be inherently miscalibrated, OOS samples, which the model cannot extrapolate to, can be associated with moderate uncertainties. On the other hand, the model could reasonably generalize to OOD samples that are flagged with high uncertainties. Hence, we define this regime as the collection of (moderate uncertainty, high MNC) and (high uncertainty, low/moderate MNC) samples;

**High Risk (■):** Finally, when both the uncertainty and non-conformity scores are high, there is no evidence that the model will behave predictably on those samples. In practice, this can correspond to both OOS and OOD samples.

### 3.3 Evaluation Metrics

While the authors of (Lahlou et al., 2023) reported the Spearman correlation between the true risk and the predicted risk on a held-out test set, DataSUITE measured the average error in top inconsistent samples. Unfortunately, neither of these metrics comprehensively indicate the behavior of failure detectors in different risk regimes. Hence, we utilize the following metrics (see Figure 3):

**False Negatives (FN)($\downarrow$)** This is the most important metric in applications, where the cost of missing to detect high risk failures is high. Hence, we measure the ratio of samples in the ID or Low Risk regimes that actually have high true risk (top $20^{th}$ percentile of all test samples).

**False Negatives (FP)($\downarrow$)** This reflects the penalty for scenarios where arbitrarily flagging harmless samples as failures. Here, we measure the ratio of samples in the Moderate or High Risk regimes that actually have low true risk (bottom $20^{th}$ percentile of all test samples).

**Confusion in Low Risk Regimes ($C_{low}$)($\downarrow$)** A common challenge in fine-grained sample grouping (ID vs Low Risk) is that detection score can confuse samples between neighboring regimes. We define this metric to measure the ratio between the $90^{th}$ percentile of the ID regime and the $10^{th}$ percentile of the Low Risk regime.

**Confusion in High Risk Regimes ($C_{high}$)($\downarrow$)** This is similar to the previous case and instead measures the confusion between the Moderate Risk and High Risk regimes.

## 4 Experiments

**Datasets.** We evaluate our framework on various datasets to demonstrate its effectiveness in identifying risk regimes. The datasets used are as follows:

1. 1D Benchmark Functions: For evaluating the performance of PAGER, we used the following standard black-box functions:

   (a) $f_1(x) = \begin{cases} x^2 & \text{if x} < 2.25 \text{ or } x > 3.01 \\ x^2 - 20 & \text{otherwise} \end{cases}$ (Figure 1)

   (b) $f_2(x) = \sin(2\pi x), x \in [-0.5, 2.5]$
   (c) $f_3(x) = a \exp(-bx) + \exp(\cos(cx)) - a - \exp(1), x \in [-5, 5], a = 20, b = 0.2, c = 2\pi$
   (d) $f_4(x) = \sin(x)\cos(5x)\cos(22x), x \in [-1, 2]$

   In each of these functions, we used 200 test samples drawn from an uniform grid and computed the evaluation metrics.

2. HD Regression Benchmarks: We also considered a set of regression datasets comprising different domains and varying dimensionality. (a) Camel (2D), (b) Levy (2D) (ben) characterized by multiple local and global minima, (c) Airfoil (5D), (d) NO2 (7D), (e) Kinematics (8D), (f) Puma (8D) (del) which are simulated datasets of the forward dynamics of different robotic control arms, (g) Boston Housing (13D) (bh), (h) Ailerons (39D) (ail) which is a dataset for predicting control action of the ailerons of an F16 aircraft, and (i) Drug-Target Interactions (32000D). For each benchmark, we created two variants: Gaps (training exposed to data with targets between $(0 - 30^{th})$ and $(60 - 100^{th})$ percentiles) and Tails (training exposed to $(0 - 70^{th})$ percentiles of

Table 1: **Metrics for 1D Benchmarks.** We report the FN, FP, $C_{low}$ and $C_{high}$ metrics on evaluation data across the entire target regime (lower the better). Note that for every metric, we identify the first and second best approach across the different benchmarks.

| Metric | Method | $f_1(x)$ | $f_2(x)$ | $f_3(x)$ | $f_4(x)$ | Metric | Method | $f_1(x)$ | $f_2(x)$ | $f_3(x)$ | $f_4(x)$ |
|---|---|---|---|---|---|---|---|---|---|---|---|
| FN↓ | DEUP | 6.19 | 6.56 | 16.57 | 27.13 | $C_{low}$↓ | DEUP | 65.90 | 57.86 | 34.13 | 169.54 |
| | DataSUITE | 14.03 | 8.8 | 16.31 | 7.2 | | DataSUITE | 59.42 | 24.61 | 22.44 | 89.51 |
| | MNC-only | 6.19 | 2.26 | 13.73 | 8.84 | | MNC-only | 57.54 | 40.1 | 31.66 | 52.24 |
| | Anchor UQ-only | 5.95 | 5.37 | 14.49 | 11.80 | | Anchor UQ-only | 40.7 | 19.88 | 25.59 | 98.92 |
| | PAGER (Score$_1$) | 5.61 | 0.0 | 11.63 | 2.40 | | PAGER (Score$_1$) | 28.08 | 7.19 | 19.94 | 12.05 |
| | PAGER (Score$_2$) | 4.79 | 5.59 | 8.43 | 5.59 | | PAGER (Score$_2$) | 20.61 | 17.82 | 16.57 | 19.74 |
| FP↓ | DEUP | 8.91 | 3.41 | 8.54 | 9.09 | $C_{high}$↓ | DEUP | 91.64 | 4.47 | 59.46 | 16.56 |
| | DataSUITE | 18.67 | 15.97 | 19.96 | 5.33 | | DataSUITE | 3.66 | 46.02 | 58.32 | 6.81 |
| | MNC-only | 9.93 | 10.42 | 8.82 | 12.18 | | MNC-only | 33.09 | 18.85 | 29.98 | 20.31 |
| | Anchor UQ-only | 5.05 | 4.93 | 6.54 | 6.01 | | Anchor UQ-only | 36.05 | 7.75 | 17.92 | 11.56 |
| | PAGER (Score$_1$) | 2.67 | 0.0 | 4.67 | 6.67 | | PAGER (Score$_1$) | 3.09 | 3.43 | 8.78 | 6.88 |
| | PAGER (Score$_2$) | 1.33 | 2.67 | 4.33 | 4.00 | | PAGER (Score$_2$) | 3.09 | 4.67 | 10.99 | 5.71 |

Table 2: **Comparison for runtime** for different failure characterization approaches.

| | Data Suite | DEUP | PAGER (Score 1) | PAGER (Score 2) |
|---|---|---|---|---|
| Runtime (sec) for 1000 samples with a single GPU | 29.8 | 18.2 | 1.55 | 40.9 |

the targets). In addition, we also considered the Skillcraft dataset , which represents real-world distribution shifts arising from change in the league index. Details about these benchmarks can be found in Appendix A.3 of the supplementary material.

3. Image Regression: We used three image regression benchmarks namely chair (yaw) angle, cell count and CIFAR-10 rotation prediction respectively. In each case, we synthesized two different variants – tails and gaps in the target variable, similar the HD regression experiments. The range of target values used in each of the experiments can be found in Figure 4.

**Baselines**. (i) DEUP (Lahlou et al., 2023) is the state-of-the-art epistemic uncertainty estimator of deep models. It utilizes a post-hoc, auxiliary error predictor that learns to predict the risk of the underlying model which is considered as a surrogate for uncertainties; (ii) DataSUITE (Seedat et al., 2022) is a task-agnostic approach that estimates the inconsistencies in the data regimes in order to assess data quality. Both the baselines however rely on the use of additional, curated calibration data to either train the error predictor in case of DEUP, or to obtain non-conformity scores that assess the sample level quality in the latter.

**Training Protocols**. For all experiments reported in this paper, we adopt the open-source $\Delta-$UQ codebase (Thiagarajan et al., 2022). With experiments on the tabular data, we use an MLP (Bishop & Nasrabadi, 2007) with 4 layers each with a hidden dimension of 128. While we used the WideResNet40-2 model (Zagoruyko & Komodakis, 2016) for the first two datasets, in the case of CIFAR-10, we randomly applied a rotation transformation [0 - 90 degrees] to each 32x32x3 image and trained a ResNet-34 model to predict the angle of rotation. The corresponding performance evaluations were carried out using the held-out test sets (e.g., 10K randomly rotated images for CIFAR-10). Without loss of generality, we utilize the $L_1$ objective for training the models. We provide the implementation details along with the hyper-parameters adopted in Appendix A.3.

## 5 MAIN FINDINGS & DISCUSSION

**Results on 1D benchmarks.** To identify different risk regimes, it is crucial for a method to align well with the training distribution (ID) and progressively flag regions of low, moderate and high risk as we move away from the inferred data manifold. From the results in Table 1 for standard 1D benchmark functions, PAGER achieves this objective effectively. Across all the metrics, our approach provides significant gains over DEUP and DataSUITE baselines. Furthermore, from the illustration in Figure 1, we observe that PAGER accurately identifies the training data regimes (Green) as part of the ID. As we traverse further from the training manifold, PAGER assigns low risk (Yellow)

Table 3: **Assessing the identified risk regimes for regression benchmarks (Gaps) with dimensionality ranging between** $2$ **and** $32{,}000$**.** We report the FN, FP, $C_{low}$ and $C_{high}$ metrics on evaluation data across the entire target regime (lower the better). Note that for every metric, we identify the first and second best approach across the different benchmarks.

| Metrics | Method | Camel | Levy | Airfoil | NO2 | Kinematics | Puma | Housing | Ailerons | DTI |
|---|---|---|---|---|---|---|---|---|---|---|
| $FN\downarrow$ | DEUP | 15.79 | 9.25 | 8.81 | 2.27 | 17.58 | 13.21 | 11.46 | 14.39 | 16.51 |
| | DataSUITE | 21.74 | 19.69 | 5.95 | 6.58 | 18.40 | 16.77 | 17.71 | 11.23 | 29.18 |
| | PAGER ($Score_1$) | 12.15 | 10.86 | 0.75 | 0.0 | 6.42 | 10.37 | 6.25 | 0.91 | 9.26 |
| | PAGER ($Score_2$) | 11.39 | 10.65 | 1.04 | 0.93 | 6.38 | 10.84 | 7.29 | 1.20 | 10.11 |
| $FP\downarrow$ | DEUP | 17.48 | 10.04 | 6.24 | 11.79 | 18.67 | 12.05 | 10.34 | 15.96 | 19.73 |
| | DataSUITE | 15.74 | 15.32 | 6.35 | 18.33 | 10.67 | 17.33 | 12.07 | 8.03 | 30.93 |
| | PAGER ($Score_1$) | 3.36 | 5.04 | 3.56 | 4.18 | 12.04 | 9.67 | 8.62 | 4.05 | 9.94 |
| | PAGER ($Score_2$) | 7.56 | 4.18 | 3.82 | 3.05 | 10.67 | 8.83 | 9.07 | 1.33 | 10.29 |
| $C_{low}\downarrow$ | DEUP | 50.59 | 34.67 | 28.23 | 19.32 | 10.71 | 14.82 | 13.86 | 15.55 | 5.46 |
| | DataSUITE | 42.92 | 71.06 | 37.11 | 47.6 | 21.96 | 15.26 | 14.8 | 30.78 | 12.8 |
| | PAGER ($Score_1$) | 14.05 | 13.62 | 11.8 | 7.01 | 12.91 | 12.44 | 13.33 | 12.90 | 2.56 |
| | PAGER ($Score_2$) | 10.13 | 10.41 | 9.93 | 6.15 | 10.93 | 8.71 | 10.42 | 11.18 | 2.83 |
| $C_{high}\downarrow$ | DEUP | 15.47 | 12.42 | 17.99 | 7.71 | 11.28 | 6.18 | 3.36 | 23.94 | 10.05 |
| | DataSUITE | 37.51 | 36.55 | 14.85 | 6.82 | 5.97 | 10.57 | 22.56 | 4.23 | 18.93 |
| | PAGER ($Score_1$) | 8.89 | 10.39 | 4.72 | 4.12 | 7.71 | 8.09 | 3.19 | 1.69 | 5.22 |
| | PAGER ($Score_2$) | 11.03 | 9.37 | 3.90 | 2.83 | 7.01 | 7.30 | 2.95 | 1.65 | 4.19 |

Table 4: **Assessing the identified risk regimes for regression benchmarks (Tails) with dimensionality ranging between** $2$ **and** $32{,}000$**.** For every metric, we identify the first and second best approach across the different benchmarks.

| Metrics | Method | Camel | Levy | Airfoil | NO2 | Kinematics | Puma | Housing | Ailerons | DTI |
|---|---|---|---|---|---|---|---|---|---|---|
| $FN\downarrow$ | DEUP | 10.53 | 7.34 | 11.28 | 13.76 | 14.39 | 16.82 | 2.11 | 18.37 | 19.23 |
| | Data SUITE | 3.84 | 9.21 | 11.02 | 12.16 | 17.59 | 22.38 | 17.89 | 17.58 | 20.06 |
| | PAGER ($Score_1$) | 0.0 | 4.56 | 1.94 | 3.65 | 8.02 | 8.78 | 1.05 | 9.59 | 6.67 |
| | PAGER ($Score_2$) | 0.25 | 4.82 | 2.48 | 3.25 | 7.18 | 10.38 | 2.32 | 9.59 | 7.13 |
| $FP\downarrow$ | DEUP | 9.53 | 7.35 | 10.82 | 9.11 | 13.02 | 14.67 | 8.77 | 12.01 | 17.34 |
| | Data SUITE | 3.83 | 6.38 | 9.15 | 9.75 | 24.0 | 26.67 | 19.3 | 12.0 | 14.07 |
| | PAGER ($Score_1$) | 0.42 | 1.68 | 2.85 | 4.27 | 6.33 | 13.33 | 3.51 | 0.80 | 9.09 |
| | PAGER ($Score_2$) | 1.68 | 2.52 | 4.29 | 6.18 | 6.18 | 12.23 | 4.26 | 0.38 | 8.36 |
| $C_{low}\downarrow$ | DEUP | 34.04 | 52.74 | 29.11 | 16.95 | 6.36 | 5.37 | 13.0 | 11.07 | 48.25 |
| | Data SUITE | 42.08 | 81.06 | 57.01 | 33.47 | 7.34 | 5.67 | 17.73 | 16.52 | 90.11 |
| | PAGER ($Score_1$) | 15.59 | 26.44 | 8.25 | 15.09 | 6.58 | 4.61 | 5.14 | 17.19 | 19.94 |
| | PAGER ($Score_2$) | 14.37 | 14.04 | 10.08 | 11.73 | 5.73 | 5.5 | 6.67 | 11.38 | 17.01 |
| $C_{high}\downarrow$ | DEUP | 23.69 | 20.75 | 14.56 | 27.34 | 6.83 | 2.63 | 5.69 | 7.25 | 39.94 |
| | Data SUITE | 17.49 | 27.32 | 18.09 | 31.58 | 10.08 | 6.41 | 5.15 | 4.97 | 64.48 |
| | PAGER ($Score_1$) | 7.5 | 17.93 | 15.19 | 12.08 | 7.14 | 2.46 | 5.07 | 2.31 | 13.35 |
| | PAGER ($Score_2$) | 6.7 | 15.18 | 16.64 | 10.68 | 7.09 | 2.81 | 4.05 | 2.43 | 11.06 |

to unseen examples that are close to the training data. Notably, as we encounter samples that are significantly out-of-sample or out-of-distribution, it consistently flags them as Moderate or High risk. Importantly, PAGER ensures a well-calibrated transition between risk regimes across the entire input space. As an ablation study, we benchmarked the failure detection performance using the (a) MNC and (b) uncertainties scores from PAGER, in order to understand the performance of each of those components independently. We include these ablation results on the 1D benchmarks along with other methods in Table 1. As expected, we find that PAGER provides consistently superior results across all datasets. The details of this study can be found in Appendix A.4.

In addition to the fidelity metrics, computational efficiency is another important aspect of failure detectors in practice. Table 2 provides the inference run-times for the different methods measured using a test set of 1000 samples (1D benchmarks). DataSUITE involves training an autoencoder followed by conformalization, while DEUP requires an auxiliary risk estimator to evaluate risk. In comparison, computing $Score_1$ with PAGER is very efficient as it basically involves only forward passes with the anchored model. $Score_2$ on the other hand requires a test-time optimization guided by the manifold regularization objective. While $Score_2$ comes with an increased computational cost, we find that it helps in resolving regimes of moderate and high risk better.

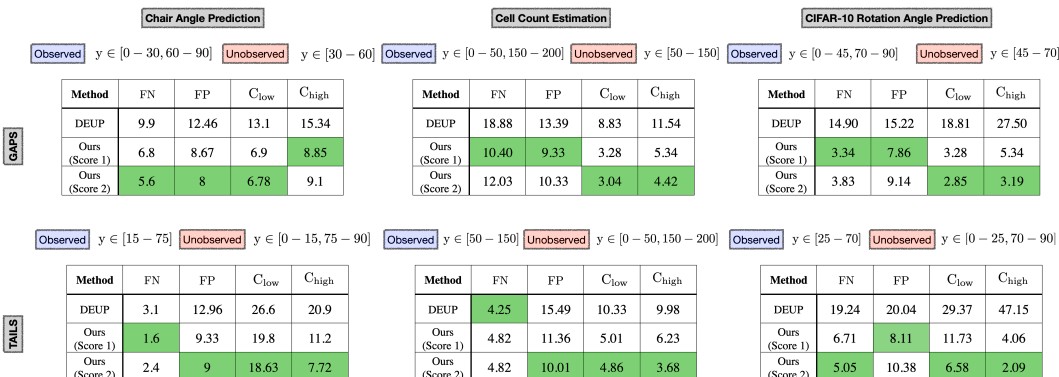

Figure 4: **Efficacy of PAGER on Image Regression Benchmarks.** We can observe that in comparison to the state-of-the art baseline DEUP, PAGER effectively minimizes the FN, FP and confusion metrics even under challenging extrapolation scenarios. We find that PAGER can consistently flag samples from the unobserved regimes which corresponds to highly erroneous predictions.

**Results on HD regression datasets.** As discussed in Section 3, it is vital to ensure that regimes that have been identified as ID or Low Risk do not correspond to large prediction errors and vice-versa thereby reducing FN and FP. From the results for suite of HD regression benchmarks, we find that PAGER consistently produces lower false negatives and false positives in comparison to the state-of-the-art baselines. This highlights the limitations of relying solely on predictive uncertainties, such as DEUP, for failure characterization. Additionally, utilizing uncertainty methods such as DataSUITE that assess data quality without task-specific considerations may not accurately identify risk regimes. Remarkably, even in higher dimensions and more complex extrapolation scenarios (e.g., Gaps and Tails, as discussed in Section 4), PAGER effectively outperform the baselines. The results presented in Tables 3 and Table 4 demonstrate the effectiveness of PAGER, showcasing an average reduction of $> 50\%$ in FN and FP scores over the baselines. Furthermore, it can be observed from our results that PAGER can significantly reduce the amount of overlap ($C_{low}$ and $C_{high}$) between the risk regime. The baselines on the other hand produce significantly higher confusion scores demonstrating their limitations in risk stratification. This observation persists even on the Skillcraft dataset characterized by real-world distribution shifts (Table 5, Appendix A.5). Finally, we notice that, despite the increased computational complexity, Score$_2$ leads to lower confusion scores compared to Score$_1$ while producing comparable FP and FN metrics (Please refer to Table 6, Appendix A.5 for additional results that demonstrate the benefits of Score$_2$).

**Results on imaging benchmarks.** Our analysis in Figure 4 reveals that our framework achieves lower FN, FP, and confusion scores compared to the baseline methods, even when confronted with challenging extrapolation regimes in imaging datasets. This demonstrates the effectiveness of our approach in handling diverse modalities of data. Additionally, we provide sample images that were accurately identified as high risk by PAGER in Appendix A.5 of the supplementary material. Notably, these examples correspond to regimes that were not encountered during training.

## 6 CONCLUSIONS

In this paper, we proposed PAGER, a framework for failure characterization in deep regression models. It leverages the principle of anchoring to integrate epistemic uncertainties and novel non-conformity scores, enabling the organization of samples into different risk regimes and facilitating a comprehensive analysis of model errors. We identify two key impacts. First, PAGER can enhance the safety of AI model deployment by proactively and preemptively detect failure cases in various high impact scenarios such as scientific simulations. This can prevent costly errors and mitigate risks associated with inaccurate predictions. Second, PAGER contributes to advancing research in failure characterization for deep regression. While we believe that it can improve reliability, its deployment and usage should be accompanied by ethical considerations and human oversight. Additional discussion on future extensions for this work can be found in Appendix A.7.

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
