## A  APPENDIX - PAGER: A FRAMEWORK FOR FAILURE ANALYSIS OF DEEP REGRESSION MODELS

### A.1  DETAILED DESCRIPTION OF ANCHORING IN PAGER

PAGER expands on the recent successes in anchoring Thiagarajan et al. (2022); Netanyahu et al. (2023) by building upon the $\Delta$-UQ methodology introduced in Thiagarajan et al. (2022). This methodology is used to estimate prediction uncertainties, which play a vital role in characterizing model risk regimes, as depicted in Figure 2 of the main paper. With that context, we now provide a concise overview of $\Delta$−UQ, its training and uncertainty estimation.

Overview: $\Delta$−UQ, short for $\Delta$−Uncertainty Quantification, is a highly efficient strategy for estimating predictive uncertainties that leverages anchoring. It belongs to the category of methods that estimate uncertainties using a single model Van Amersfoort et al. (2020); Liu et al. (2020). $\Delta$−UQ has been demonstrated to be an improved and scalable alternative to Deep Ensembles Lakshminarayanan et al. (2017), eliminating the need to train multiple independent models for estimating uncertainties. The core idea behind $\Delta$−UQ is based on the observation that the injection of constant biases (anchors) to the input dataset produces different model predictions as a function of the bias. To that end, models trained using the same dataset but shifted by respective biases generates diverse predictions. This phenomenon arises from the fact that the neural tangent kernel (NTK)Jacot et al. (2018) induced in deep models lacks invariance to input data shifts Bishop & Nasrabadi (2007). Consequently, the variance among these models *a.k.a anchored ensembles* serves as a strong indicator of predictive uncertainty. Based on this observation, $\Delta$−UQ follows a simple strategy to consolidate the anchored ensembles into a single model training, where the input is reparameterized as an anchored tuple, as described in Section 2 of the main paper. It is important to note that $\Delta$−UQ performs anchoring in the input space for both vector-valued and image data.

Training: In this phase, for every training pair $\{x, y\}$ drawn from the dataset $\mathcal{D}$, a random anchor $r_k$ is selected from the same training dataset. Both the input $x$ and the anchor $r_k$ are transformed into a tuple given as $[r_k, x - r_k]$. Importantly, this reparameterization does not alter the original predictive task, but instead of using only $x$, the tuple $[r_k, x - r_k]$ is mapped to the target $y$. For vector-valued data, $\Delta$−UQ constructs the tuples by concatenating the anchor $r_k$ and the residual along the dimension axis. In the case of images, the tuples are created by appending along the channel axis, resulting in a $6$−channel tensor for each 3-channel image. These tuples are organized into batches and used to train the models. Throughout the training process, in expectation, every sample $x$ is anchored with every other sample in the dataset. The goal here is that the predictions for every $x$ should remain consistent regardless of the chosen anchor. The training objective is given by:

$$\theta^* = \arg\min_{\theta} \quad \mathcal{L}(y, F_\theta([r_k, x - r_k]), \tag{4}$$

where $\mathcal{L}(.)$ is a loss function such as MAE or MSE. In effect, the $\Delta$−UQ training enforces that for every input sample $x$, $F_\theta([r_1, x - r_1]) = F_\theta([r_2, x - r_2]) = \cdots = F_\theta([r_k, x - r_k])$, where $F_\theta$ is the underlying model that operates on the tuple $([r_k, x - r_k])$ to predict $y$.

Uncertainty Estimation: During the inference phase, using the trained model with weights $\theta^*$, we compute the prediction $y_t$ for any test sample $x_t$. This is performed by averaging the predictions across $K$ randomly selected anchors drawn from the training dataset. The standard deviation of these predictions is then used as the estimate for predictive uncertainty. The equations for calculating the mean prediction and uncertainty around a sample can be found in Equation 1 of the main paper.

### A.2  ALGORITHM LISTINGS FOR PAGER

Algorithms 1,2 and 3 provide the details for estimating predictive uncertainty, non-conformity scores - $\texttt{Score}_1$ and $\texttt{Score}_2$ respectively in PAGER.

### A.3  DESCRIPTION OF OUR TRAINING PROTOCOLS

In the case of Cell Count and Chair Angle benchmarks, we train an anchored $40 - 2$ WideResnet model. The training is performed with a batch size of $128$ for 100 epochs. We utilize the ADAM optimizer with momentum parameters of $(0.9, 0.999)$ and a fixed learning rate of $1e - 4$. To train the anchored auto-encoder for computing $\texttt{Score}_2$, we employ a convolutional architecture with

---

**Algorithm 1** PAGER: Predictive Uncertainty Estimation

---

1: **Input**: Input test samples $\{x_i^t\}_{i=1}^N$, Pre-trained anchored model $F_{\theta^*}$, Anchors $\{r_k\}_{k=1}^K$ drawn from the training dataset $\mathcal{D}$
2: **Output**: Predictive Uncertainties (Unc) for $\{x_i^t\}_{i=1}^N$
3: **Initialize**: Unc = list()
4: **for** $i$ in 1 to $N$ **do**
5: $\quad \mu(y_i^t|x_i^t) = \frac{1}{K}\sum_{k=1}^K F_{\theta^*}([r_k, x_i^t - r_k]);$
6: $\quad \sigma(y_i^t|x_i^t) = \sqrt{\frac{1}{K-1}\sum_{k=1}^K (F_{\theta^*}([r_k, x_i^t - r_k]) - \mu(y_i^t|x_i^t))^2};$
7: $\quad$ Unc[i] $= \sigma(y_i^t|x_i^t)$
8: **end for**
9: **return**: Unc

---

**Algorithm 2** PAGER: $\text{Score}_1$ Computation

---

1: **Input**: Input test samples $\{x_i^t\}_{i=1}^N$, Pre-trained anchored model $F_{\theta^*}$, Train data subset $\{r_k, y_k\}_{k=1}^K$
2: **Output**: $\text{Score}_1$ for $\{x_i^t\}_{i=1}^N$
3: **Initialize**: $\text{Score}_1$ = list()
4: **for** $i$ in 1 to $N$ **do**
5: $\quad$ s $= \max\limits_{k} \left\|y_k - F_{\theta^*}([x_i^t, r_k - x_i^t])\right\|_1 \quad \forall k \in \{1\cdots K\};$
6: $\quad \text{Score}_1[i] = $ s
7: **end for**
8: **return**: $\text{Score}_1$

---

**Algorithm 3** PAGER: $\text{Score}_2$ Computation

---

1: **Input**: Input test samples $\{x_i^t\}_{i=1}^N$, Pre-trained anchored model $F_{\theta^*}$, Pre-trained anchored auto-encoder $A$, Train data subset $\{r_k, y_k\}_{k=1}^K$, Learning rate $\eta$, Weighing Factor $\lambda$, No. of iterations $T$
2: **Output**: $\text{Score}_2$ for $\{x_i^t\}_{i=1}^N$
3: **Initialize**: $\text{Score}_2$ = list()
4: **for** $i$ in 1 to $N$ **do**
5: $\quad$ **Initialize**: $\bar{x} \leftarrow x_i^t$
6: $\quad$ **for** $iter$ in 1 to $T$ **do**
7: $\qquad$ Compute $\mathcal{R}(\bar{x}) = \left\|\bar{x} - A([x_i^t, \bar{x} - x_i^t])\right\|_2 + \left\|x_i^t - A([\bar{x}, x_i^t - \bar{x}])\right\|_2.$
8: $\qquad$ Compute $L = \frac{1}{K}\sum\limits_{k}\|y_k - F_{\theta^*}([\bar{x}, r_k - \bar{x}])\|_1 + \lambda\mathcal{R}(\bar{x})$
9: $\qquad$ Update $\bar{x} \leftarrow \bar{x} - \eta\nabla_{\bar{x}}L$
10: $\quad$ **end for**
11: $\quad \text{Score}_2[i] = \|x - \bar{x}\|_2$
12: **end for**
13: **return**: $\text{Score}_2$

---

an encoder-decoder structure. The encoder consists of two convolutional layers with kernel sizes of $(3,3)$ and appropriate padding, as well as MaxPooling operations. The decoder comprises two transposed convolutional layers with stride 2 to reconstruct the input images. We train the anchored auto-encoder using a batch size of 128 for 100 epochs. The ADAM optimizer with momentum parameters $(0.9, 0.999)$, and a fixed learning rate of $1e-3$, is used for training. As mentioned in the main paper, for the case of CIFAR-10, we train a ResNet-34 model with the standard training configurations. For the other regression benchmarks, we used a standard MLP with 5 hidden layers, ReLU activation and batchnorm. They were all trained for 5000 epochs with learning rate $5e-5$ and ADAM optimizer.

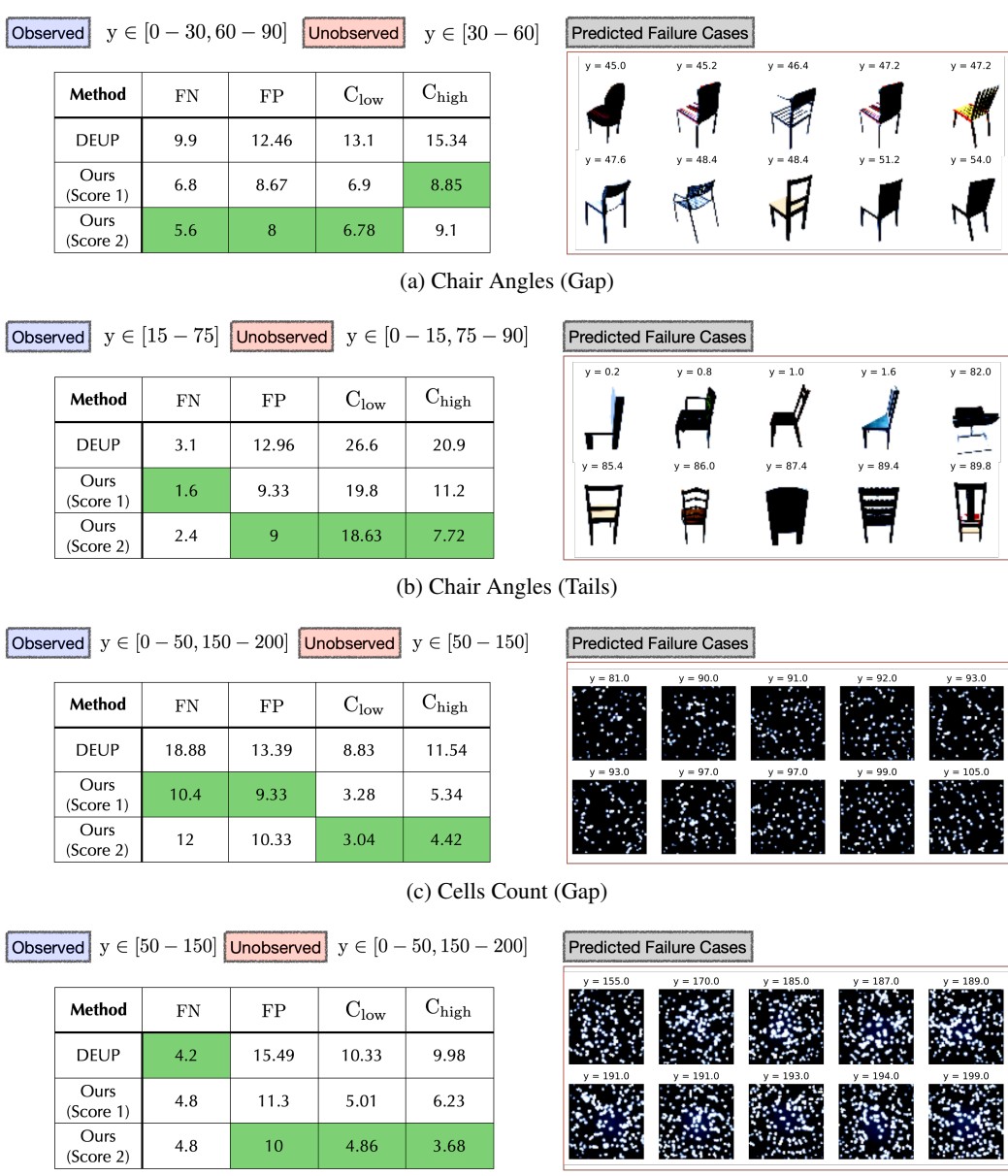

Figure 5: **Efficacy of PAGER on Image Regression Benchmarks.** We can observe that in comparison to the state-of-the art baseline DEUP, PAGER effectively minimizes the FN, FP and confusion metrics even under challenging extrapolation scenarios. We find that PAGER can consistently flag samples from the unobserved regimes which corresponds to highly erroneous predictions.

## A.4 ABLATIONS

While PAGER jointly considers both anchoring-based uncertainties and non-conformity scores to identify different risk regimes, it is important understand the performance of using each of those components independently. To this end, we create the two following baselines and report performance comparisons on the tabular benchmarks.

**NC-only.** Our NC scores (Scores 1 and 2) measure either the relative change in the target value or distances in the input space. Since those scores are unnormalized, they behave differently across different data regimes. For e.g., NC scores can be high for scenarios where epistemic uncertainties

Table 5: **Additional results with SKillcraft dataset**. While we used synthetic shifts in the HD benchmark experiments, real-world shifts were used in the cases of SkillCraft. For every metric, we identify the first and second best approach across the different benchmarks. Corroborating with our findings in the paper, naame consistently outperforms existing baselines across all four metrics.

| Metrics | Method | SkillCraft |
|---|---|---|
| FN↓ | DEUP | 7.83 |
| | Data SUITE | 13.45 |
| | PAGER ($Score_1$) | 4.43 |
| | PAGER ($Score_2$) | 6.6 |
| FP↓ | DEUP | 14.33 |
| | Data SUITE | 17.03 |
| | PAGER ($Score_1$) | 8.91 |
| | PAGER ($Score_2$) | 10.5 |
| $C_{low}$↓ | DEUP | 5.47 |
| | Data SUITE | 6.92 |
| | PAGER ($Score_1$) | 4.72 |
| | PAGER ($Score_2$) | 5.8 |
| $C_{high}$↓ | DEUP | 6.69 |
| | Data SUITE | 7.01 |
| | PAGER ($Score_1$) | 2.94 |
| | PAGER ($Score_2$) | 2.05 |

are high or low (Fig. 1 in the main paper). Hence, they are insufficient to accurately rank samples on their own. However, designing a normalization strategy that works even for unseen test data is non-trivial, and hence PAGER works with unnormalized scores only. Table **??** shows results for the 1D benchmarks used in our study. While NC-only baseline can reasonably control FP, it is not able to reduce the FN.

**Anchor-UQ.** In this baseline, we directly utilized the uncertainties from Δ-UQ to detect risk regimes. Table 1 from the main paper includes the performance assessment for this baseline as well. We observe that uncertainties from the anchored model are a much more effective baseline, even outperforming DEUP in all cases.

In comparison to both these baselines, PAGER leads to significant improvements in all metrics, emphasizing the importance of considering both uncertainty and non-conformity together.

### A.5 ADDITIONAL RESULTS

**Image regression experiment.** For the cell count and chair angle prediction benchmarks from the main paper, we provide examples of high-risk sample as detected by PAGER. Please refer to Figure 5 for the examples. Notably, these samples correspond to regimes that were not encountered during training.

**Skillcraft dataset.** In Table 5, we report the results for the Skillcraft benchmark where the evaluation data belongs to unobserved league indices (represents real-world distribution shifts). As expected, PAGER provides considerable improvements across all four metrics.

**Demonstrating benefits of** `Score_2`**.** Conceptually, `Score_2` employs a more rigorous optimization approach to effectively address test scenarios in which both a test sample and its residual are individually out-of-distribution (OOD), not just their combination, with respect to $P(X)$ and $P(\Delta)$. As demonstrated earlier, `Score_2` often demonstrates noteworthy improvements in confusion scores ($C_{low}$ and $C_{high}$) in both *gaps* and *tails* settings. This is particularly valuable in practice where users cautiously identify failures (Top K%) and it is essential to prevent the misclassification of some high-risk samples as moderate risk. Another scenario where `Score_2` can be very useful is when the test samples are drawn from a different distribution compared to training (referred to as covariate shifts). To demonstrate this hypothesis, we repeated the CIFAR-10 rotation angle prediction experiment by applying natural image corruptions (defocus blur and frost) at different varying severity levels. Interestingly, we observed significant improvements in both FP and FN scores with `Score_2` as the

Table 6: **Benefits of** Score$_2$. Performance of PAGER on the CIFAR-10 rotation angle prediction model, where the test images undergo natural image corruptions. As the severity of image corruption increases, the benefits Score$_2$ become apparent in both FP and FN metrics.

| Metric | Score | Defocus Blur | | | Frost | | |
|---|---|---|---|---|---|---|---|
| | | Sev.1 | Sev. 3 | Sev. 5 | Sev.1 | Sev. 3 | Sev. 5 |
| FP↓ | Score$_1$ | 8.05 | 10.29 | 18.54 | 7.81 | 8.80 | 16.52 |
| | Score$_2$ | 9.28 | 9.95 | 14.08 | 9.06 | 9.58 | 11.76 |
| FN↓ | Score$_1$ | 3.65 | 5.85 | 19.81 | 3.77 | 6.12 | 15.63 |
| | Score$_2$ | 3.92 | 5.05 | 14.99 | 4.01 | 5.68 | 11.14 |

Table 7: **Implementing PAGER with an anchored regression head**. Even with only an anchored regression head on top of a pre-trained feature extractor, PAGER performs effectively in terms of recovering the different risk regimes and is consistently superior to baselines.

| Metrics | Method | CIFAR-10 |
|---|---|---|
| FN↓ | DEUP | 14.9 |
| | PAGER (Full Anchoring) | 3.34 |
| | PAGER (Anchored Head) | 4.78 |
| FP↓ | DEUP | 15.22 |
| | PAGER (Full Anchoring) | 7.86 |
| | PAGER (Anchored Head) | 8.33 |
| C$_{low}$↓ | DEUP | 18.81 |
| | PAGER (Full Anchoring) | 3.28 |
| | PAGER (Anchored Head) | 4.96 |
| C$_{high}$↓ | DEUP | 27.50 |
| | PAGER (Full Anchoring) | 5.34 |
| | PAGER (Anchored Head) | 6.05 |

severity increases. In summary, while Score$_1$ excels in scalability and is well-suited for online evaluation, Score$_2$ proves advantageous in addressing challenging testing scenarios.

## A.6 Implementing PAGER with an Anchored Regression Head

Regarding the application of PAGER to standard models (trained without anchoring), we want to begin by emphasizing that anchoring does not necessitate any adjustments to the optimizer, loss function, or training protocols, e.g., incorporation of additional strategies such as mixup. The sole modification lies in the input layer, requiring additional dimensions for vector-valued data or channels for images, and a modest addition of parameters to the first layer. This approach is adaptable to any architecture, be it a Multi-Layer Perceptron (MLP), Convolutional Neural Network (CNN), or Vision Transformer (ViT). Recent research indicates that incorporating anchoring not only aligns with test performance but can even enhance generalization to out-of-distribution (OOD) regimes. However, in situations involving pre-trained models, it is feasible to train solely an anchoring-based regression head attached to a conventionally trained feature extractor backbone, effectively implementing PAGER. Optionally, one may choose to fine-tune the feature extractor concurrently with the anchored regression head in an end-to-end manner, adhering to standard practices. As an illustration, in the experiment on CIFAR-10 rotation angle prediction under the Gaps setting with Score$_1$, we considered a variant, where we exclusively trained an anchored regression head while keeping the feature extractor frozen. Note, the feature extractor was obtained through standard training on the same dataset. Our findings reveal that even with this approach, the performance is comparable to that of a fully anchored ResNet-34 model.

## A.7 DISCUSSION: UTILITY OF PAGER IN ANALYZING CLASSIFIER MODELS

In classification tasks, incorrectly assigned labels are considered failures. Therefore, detecting failure is framed as assessing the probability of an inaccurate label prediction. State-of-the-art approaches typically employ auxiliary predictors, trained retrospectively, to correlate various sample-level scores (such as uncertainties, smoothness, and disagreement across multiple hypotheses) with the likelihood of an incorrect prediction. This approach inherently depends on extra labeled calibration data and has implications for the generalization of the learned failure detector in the face of shifts.

A key difficulty in characterizing failures in regression tasks stems from the variability in the permissible tolerance levels on error estimates, which can differ across use-cases. This is in contrast to the clear-cut definition of failure in classification tasks. Addressing this challenge, PAGER proposes to organize samples into various risk regimes rather than relying on a binary pass-fail (0-1) categorization. This approach not only allows for a nuanced assessment without the need for a predefined, rigid definition of failure, but also eliminates the necessity for labeled calibration data. It's worth noting, however, that state-of-the-art methods like DEUP, which seek to emulate failure detectors from the classification literature by fitting an auxiliary predictor to estimate the loss, exhibit poor performance in practice.

Ultimately, while the risk regime characterization approach introduced by PAGER may not find direct applicability in the existing frameworks for failure detection in deep classifiers, it's crucial to underscore that the proposed uncertainty and non-conformity scores can serve as alternatives to widely used scoring functions for training the auxiliary predictor. Nevertheless, unlike the implementation in PAGER, adopting these scores in current classifier failure detectors would necessitate access to additional calibration data. This naturally emphasizes the importance of developing calibration-free failure detectors, even in the context of multi-class classification, and we reserve that for future work.