# OpenReview forum: "PAGER: A Framework for Failure Analysis of Deep Regression Models"
_ICLR.cc/2024/Conference — Submitted to ICLR 2024_

### Official Review · Reviewer_Kt6W · 2023-10-31

**Soundness:** 3 good
**Presentation:** 3 good
**Contribution:** 3 good
**Rating:** 6
**Confidence:** 2

**Summary:**

This paper presents a risk estimation methods that can categorize regression prediction failure into four groups, including ID, Low Risk, Moderate Risk, and High Risk. The paper starts with demonstrating the weakness of epistemic uncertainty based method and show that it is insufficient to evaluate predictive risk when there is Out-of-Support in the modelling. With such motivation, the paper presents its method based on anchoring  predictive model. The transformation of inputs is able to construct a connection between training data points with test data point such that OOS / OOD data could be flagged out through carefully defined functions. In experiments, the paper includes synthetic benchmark function as well as real regression datasets to show the effect of the proposed method. Baselines are those epistemic uncertainty based solutions. Evaluation metrics are somewhat not standard, which is probably introduced by this paper itself. Thus, I cannot tell if the experimental result is indeed trustworthy.

**Strengths:**

1. The paper focus on the risk estimation of regression tasks, which is less explored compared to classification tasks. While uncertainty estimation based solution has been there for a long time, there were less contribution made in this field that is not based on uncertainty estimation. This makes the paper stand out as a novel contribution.
2. The paper is fairly well written with sufficient evidence to support its claim. While some concepts may be a bit hypothetical (I mean not really possible to distinguish between OOD OOS in practice), the motivation and solution are well connected.

**Weaknesses:**

1. The solution is based on anchoring predictive model. I realized the novelty majorly comes from the nature of the anchoring predictive model, which makes me concerning the contribution this particular paper (as it is more incremental now). In addition, how does people use this method to measure regression risk if they don't use anchoring model (which more likely happen in real world)?
2. Multiple concepts introduced in this paper are hypothetical and may not be possible to know in practice. E.g. OOS / OOD are hard to distinguish.
3. The evaluation metric introduced in this paper is hard to be convincing in terms of the setting with 20 percentile or 90 percentile in the False Positive and $C_{low}$. Why? What makes the authors to choose those numbers. Is that the real reason of better performance compared with existing solutions?

Minors:
Type: Section 3.3, False Negative (FP) should be False Positive.

**Questions:**

My questions are included in the weaknesses section.

---

> ### Author Response · Authors · 2023-11-11
> **Author Response to Reviewer Kt6W**
>
> We thank the reviewer for the positive feedback and for raising important questions. Here is our detailed response. If our responses have adequately addressed the concerns, we request the reviewer to increase their rating to help champion our paper.
>
> > **Novelty and Implications of using Anchoring**
>
> While our solution is rooted in the established anchoring framework, this paper introduces several noteworthy contributions.
> + We uncover a crucial insight that **uncertainties alone are insufficient** for a comprehensive characterization of failures in regression models.
> + We advocate for incorporating **manifold non-conformity**, emphasizing adherence to the joint data distribution, as a valuable complement to uncertainty estimates.
> + In an extension beyond the existing anchoring framework, we present a surprising finding that **non-conformity can be achieved through reverse anchoring** without the need for auxiliary models.
> + Additionally, we propose a flexible analysis of model errors by introducing the concept of **risk regimes**, avoiding the necessity for a rigid definition of failure (such as an incorrect label) and the need for additional calibration data commonly seen in classification problems.
>
> Concerning the utilization of anchored models as the foundational framework, we believe that it is neither a weakness nor a limitation of PAGER. In this regard, we would like to offer two practical observations:
>
> + **Anchoring does not necessitate any adjustments to the optimizer, loss function, or training protocols**. The sole modification lies in the input layer, requiring additional dimensions for vector-valued data or channels for images, and a modest addition of parameters to the first layer. This approach is adaptable to any architecture, be it a Multi-Layer Perceptron (MLP), Convolutional Neural Network (CNN), or Vision Transformer (ViT). Recent research indicates that **incorporating anchoring not only aligns with test performance but can even enhance generalization to out-of-distribution (OOD) regimes** [1,2].
> + Even in situations involving pre-trained models, it is feasible to **train solely an anchoring-based regression head attached to a conventionally trained feature extractor backbone**, effectively implementing PAGER. Optionally, one may choose to fine-tune the feature extractor concurrently with the anchored regression head in an end-to-end manner, adhering to standard practices.
>
> As an illustration, in the experiment on CIFAR-10 rotation angle prediction under the Gaps setting with $\mathtt{Score1}$, we considered a variant, where we exclusively trained an anchored regression head while keeping the feature extractor frozen. Note, the feature extractor was obtained through standard training on the same dataset. Our findings reveal that even with this approach, the **performance is comparable to that of a fully anchored ResNet-34 model**. We have updated the paper with this result (Appendix A.6)
>
> | Method                          | FN       | FP       | $C_{\text{low}}$ | $C_{\text{high}}$ |
> |---------------------------------|----------|----------|------------------|-------------------|
> | DEUP                            |   14.9   |   15.22  |       18.81      |       27.50       |
> | Full Anchored Training          | **3.34** | **7.86** |     **3.28**     |      **5.34**     |
> | Anchored Regression Head (only) |  _4.78_  |  _8.33_  |      _4.96_      |       _6.05_      |
>
>
> > **Definitions for OOD and OOS**
>
> Thank you for your feedback. Our definitions of OOD (Out of Distribution) and OOS (Out of Support) are directly derived from existing literature [2], where the characterization is rooted in various regimes from which the test samples originate. Although an alternative variation, Out-of-Combination (OOC), has also been considered, in our framework, we categorize it within the broader umbrella of Out-of-Support data. Alongside these definitions, we have incorporated Figure 2 to offer readers a visual representation for enhanced clarity.

---

> > ### Author Response · Authors · 2023-11-11
> > **Author Response (contd.)**
> >
> > > **Clarification on evaluation metrics reported in the paper**
> >
> > The defined thresholds in PAGER  **merely serve as guidelines for its usage, and the reported improvements do NOT hinge on them**. Addressing the inherent difficulty in characterizing failures in regression settings, where acceptable error tolerance can significantly vary across applications, it is a common practice to flag the most confident and error-prone samples. However, PAGER for the first time offers a nuanced characterization through multiple distinct risk regimes for ranking samples.
> >
> > **Regarding Clow and Chigh**: The selection of the 90th percentile and 10th percentile to gauge the error ratio between moderate and high-risk bins (or equivalently ID and low-risk bins) is **intentionally more conservative and challenging**. Even when the thresholds are relaxed, the performance gap persists in PAGER, as the average error in the high-risk bin increases, and that of the moderate-risk bin decreases, resulting in a lower ratio. An illustrative example of Chigh on the CIFAR-10 rotation angle prediction task with various thresholds is provided.
> >
> > |      Method     | 90th per. of Moderate Risk / 10th pct. of High Risk | 80th per. of Moderate Risk / 20th pct. of High Risk | 70th per. of Moderate Risk / 30th pct. of High Risk |
> > |:---------------:|:---------------------------------------------------:|:---------------------------------------------------:|:---------------------------------------------------:|
> > |       DEUP      |                        27.50                        |                        22.95                        |                        17.53                        |
> > | PAGER (Score 1) |                       **5.34**                      |                       **2.58**                      |                       **1.61**                      |
> >
> > **False Positives (FP) and False Negatives (FN)**: The utilization of the 20th and 80th percentiles as thresholds for assessing FP and FN align with established protocols for evaluating failure detectors [3]. **While previous studies often compared the average Mean Squared Error (MSE) of samples in the bins, we perform a more stringent evaluation** by quantifying the rates of false positives and negatives.
> >
> > In conclusion, although the selected thresholds are provided as guidelines and can be adjusted according to the specific requirements of an application, we advocate for a more rigorous evaluation and, in fact, present results for challenging scenarios.
> >
> > > **References**
> >
> > + [1] Single model uncertainty estimation via stochastic data centering, NeurIPS 2022.
> > + [2] Learning to extrapolate: A transductive approach, ICLR 2023.
> > + [3] Data-SUITE: Data-centric identification of in-distribution incongruous examples, ICML 2022.

---

> > > ### Comment · Reviewer_Kt6W · 2023-11-22
> > >
> > > Thank you for the response. I will keep my current score.

---

> > > > ### Author Response · Authors · 2023-11-22
> > > > **Thank you!**
> > > >
> > > > Thank you for checking our response. We wanted to check with you if you had any additional questions that we could answer before the discussion phase ends. Also, if you find our responses satisfactory, will you be open to raising the score to help champion our paper?

---

### Official Review · Reviewer_HtuK · 2023-10-31

**Soundness:** 3 good
**Presentation:** 3 good
**Contribution:** 3 good
**Rating:** 6
**Confidence:** 2

**Summary:**

In this paper, the author provide a new framework for failure mode analysis in regression problem where they can identify different regions in the input space based on the model's generalization. They use the idea of anchoring and provide uncertainty and non-conformity scores to test samples. Based on these two scores, they categorize samples. They further evaluate their method and compare it to existing methods based on FN in detecting high-risk samples, FP in detecting low-risk samples, and confusion error in both low and high-risk regions. Their method brings improvement over the existing results.

**Strengths:**

+ The observation that an uncertainty score is insufficient and a complementary non-conformity score can better organize different regions is interesting.
+ The idea of using anchors to obtain uncertainty and non-conformity is interesting.
+ Evaluation is comprehensive and includes various methods on different datasets.
+ Propose method ($\text{score}_1$) is efficient and is faster than existing work in inference time.

**Weaknesses:**

+ $\text{score}_2$ is slower than $\text{score}_1$ in inference time, but it is not significantly better than $\text{score}_1$ in all settings; I'd expect to get more information about why this method should be used and what settings it performs better.

**Questions:**

i'd like to get more insight about the usefulness of $\text{score}_2$.

---

> ### Author Response · Authors · 2023-11-11
> **Author Response to Reviewer HtuK**
>
> We thank the reviewer for their positive assessment of our work.  If our responses have adequately addressed the concerns, we request the reviewer to increase their rating to help champion our paper.
>
> > **Utility of $\mathtt{Score2}$ and Comparison to $\mathtt{Score1}$**
>
> Conceptually, $\mathtt{Score2}$ employs a more rigorous optimization approach to **effectively address test scenarios in which both a test sample and its residual are individually out-of-distribution (OOD), not just their combination, with respect to $P(X)$ and $P(\Delta)$**. While the computationally efficient $\mathtt{Score1}$ tends to perform comparably (or even better) to $\mathtt{Score2}$ under challenging extrapolation conditions (such as gaps and tails in the target variable), $\mathtt{Score2}$ often demonstrates noteworthy enhancements in confusion scores (Clow and Chigh). This becomes particularly valuable in practical applications **where users cautiously identify failures (Top K%) and it is essential to prevent the misclassification of some high-risk samples as moderate risk**.
>
> Another scenario where $\mathtt{Score2}$ can be very **useful is when the test samples are drawn from a different distribution compared to training (referred to as covariate shifts)**. To demonstrate this hypothesis, we repeated the CIFAR-10 rotation angle prediction experiment by applying *natural image corruptions (defocus blur and frost) at different severity levels*. Interestingly, we observed significant improvements in both false positive (FP) and false negative (FN) scores with $\mathtt{Score2}$ as the severity of corruption increased. In summary, while $\mathtt{Score1}$ excels in scalability and is well-suited for online evaluation, $\mathtt{Score2}$ proves advantageous in addressing challenging testing scenarios. We have updated the paper with these results (Appendix A.5).
>
> |            	|                   	| **Defocus Blur** 	| **Defocus Blur** 	| **Defocus Blur** 	|  **Frost** 	|  **Frost** 	|  **Frost** 	|
> |---------------|-----------------------|-------------------|-------------------|-------------------|---------------|---------------|---------------|
> | **Metric** 	|     **Score**     	|    **Sev. 1**    	|    **Sev. 3**    	|    **Sev. 5**    	| **Sev. 1** 	| **Sev. 3** 	| **Sev. 5** 	|
> |     FP     	| $\mathtt{Score1}$ 	|     **8.05**     	|       10.29      	|       18.54      	|  **7.81**  	|  **8.80**  	|    16.52   	|
> |     FP     	| $\mathtt{Score2}$ 	|       9.28       	|     **9.95**     	|     **14.08**    	|    9.06    	|    9.58    	|  **11.76** 	|
> |     FN     	| $\mathtt{Score1}$ 	|     **3.65**     	|       5.85       	|       19.81      	|    3.77    	|    6.12    	|    15.63   	|
> |     FN     	| $\mathtt{Score2}$ 	|       3.92       	|     **5.05**     	|     **14.99**    	|    4.01    	|  **5.68**  	|  **11.14** 	|

---

> > ### Comment · Reviewer_HtuK · 2023-11-21
> >
> > I appreciate the authors for providing explanations and experiments regarding $\\text{Score2}$.

---

> > > ### Author Response · Authors · 2023-11-21
> > > **Thank you!**
> > >
> > > Thank you for your feedback. We wanted to check with you if you had any additional questions that we can answer before the discussion phase ends. Also, if you find out responses satisfactory, will you be open to raising the score to help champion our paper?

---

### Official Review · Reviewer_LqcK · 2023-11-01

**Soundness:** 4 excellent
**Presentation:** 4 excellent
**Contribution:** 4 excellent
**Rating:** 6
**Confidence:** 4

**Summary:**

The paper addresses the need for proactively detecting potential prediction failures in AI regression models. The author introduces a framework called PAGER that combines epistemic uncertainties and non-conformity scores to systematically detect and characterize failures in deep regression models. PAGER is shown to effectively identify areas of accurate generalization and detect failure cases in various scenarios.

**Strengths:**

1. The presentation of the paper is clear, and it is insightful to characterize different failures of models in a unified framework.

2. The idea of unifying epistemic uncertainties and complementary non-conformity scores is reasonable.

3. The experimental results verify that PAGER achieves great improvement in failure analysis of deep regression models on synthetic and real-world benchmarks.

**Weaknesses:**

Can the framework proposed in this paper for analyzing model failures be applied to multi-class classification tasks, and what are the potential differences between it and the regression task studied in the paper?

**Questions:**

Please refer to [Weaknesses].

---

> ### Author Response · Authors · 2023-11-11
> **Author Reponse to Reviewer LqcK**
>
> We thank the reviewer for their positive feedback about the paper.  If our responses have adequately addressed the concerns, we request the reviewer to increase their rating to help champion our paper
>
> > **Utility of PAGER on Multi-class Classifier Models**
>
> Thanks for raising this important question.
>
> In classification tasks, incorrectly assigned labels are considered failures. Therefore, detecting failure is framed as **assessing the probability of an inaccurate label prediction**. State-of-the-art approaches typically employ auxiliary predictors, trained retrospectively, to correlate various sample-level scores (such as uncertainties, smoothness, and disagreement across multiple hypotheses) with the likelihood of an incorrect prediction. This approach inherently depends on extra labeled calibration data and has implications for the generalization of the learned failure detector in the face of shifts.
>
> A key difficulty in characterizing failures in regression tasks stems from the **variability in the permissible tolerance levels on error estimates**, which can differ across use-cases. This is in contrast to the clear-cut definition of failure in classification tasks. Addressing this challenge, PAGER proposes to organize samples into various risk regimes rather than relying on a binary pass-fail (0-1) categorization. This approach not only allows for a nuanced assessment without the need for a predefined, rigid definition of failure, but also eliminates the necessity for labeled calibration data. It’s worth noting, however, that state-of-the-art methods like DEUP, which seek to emulate failure detectors from the classification literature by fitting an auxiliary predictor to estimate the loss, exhibit poor performance in practice.
>
> Ultimately, while the risk regime characterization approach introduced by PAGER may not find direct applicability in the existing frameworks for failure detection in deep classifiers, it’s crucial to underscore that the proposed uncertainty and non-conformity scores can serve as alternatives to widely used scoring functions for training the auxiliary predictor. Nevertheless, unlike the implementation in PAGER, adopting these scores in current classifier failure detectors would necessitate access to additional calibration data. **We have added a discussion to the paper (Appendix A.7), emphasizing the importance of developing calibration-free failure detectors, even in the context of multi-class classification**.

---

> > ### Author Response · Authors · 2023-11-22
> > **Request**
> >
> > Can you please check our response and let us know if you had any additional questions that we could answer before the discussion phase ends. Also, if you find our responses satisfactory, will you be open to raising the score to help champion our paper?

---

### Author Response · Authors · 2023-11-20
**Request**

Dear reviewers,

As the author-reviewer discussion is nearing its final phase, we wanted to request you to check our responses and let us know if there are any additional comments or questions that we can address.

Here is a quick summary of our response:
+ Discussion on **utility of PAGER to multi-class classification models** (Reviewer LqcK)
+ An **additional experiment to demonstrate the utility of Score 2** compared to Score 1 in PAGER (Reviewer HtuK)
+ Clarification of the **novelty and implications of using anchored training** (Reviewer Kt6W)
+ An additional experiment to illustrate how **PAGER can be used even with pre-trained models** (Reviewer Kt6W)
+ Insights into the **choice of thresholds** for computing the evaluation metrics (Reviewer Kt6W)

We appreciate you for taking the time to review our work and providing useful feedback.

---

### Meta-Review · Area_Chair_znhj · 2023-12-13

**Metareview:**

This paper has been assessed by three knowledgeable reviewers who assigned it unanimously as marginally acceptable. The authors attempted to address concerns raised by th reviewers in the discussion, but the two reviewers who responded to those decided to keep their original scores. The paper proposes a fundamentally incremental novelty in the task of assessing prediction risk of deep regressors. Its novelty is in using anchoring approach for deep models. The reviewers questioned the utility of the method beyond deep regressors, and were not convinced about the non-epistemic metrics proposed by the authors. A comparison versus robustness-based metrics of risk, and relationship to the more classic statistical approaches including robust regression, could substantially enhance the story. The paper presents a promising body of work whose presentation can be further enhanced without substantial efforts.

**Justification For Why Not Higher Score:**

This paper is marginally below the ICLR acceptance threshold, but it can be improved for the subsequent submission.

**Justification For Why Not Lower Score:**

N/A

---

### Decision · Program_Chairs · 2024-01-16

Reject